# The Child–Mother–Father–Teacher Relationship Network in Kindergarten and Its Association with Children’s Social and Academic Development: An Ecological Perspective

**DOI:** 10.3390/children10071102

**Published:** 2023-06-22

**Authors:** Ifat Weisberger, Yair Ziv

**Affiliations:** Department of Counseling and Human Development, Faculty of Education, University of Haifa, Haifa 3498838, Israel; ifatweisber@gmail.com

**Keywords:** parent–teacher relationships, socioemotional development, kindergarten, ecological perspective

## Abstract

This study examines how a set of the child’s proximal relationships (mother–child, father–child, and teacher–child) and parent–teacher relationships relate to the child’s prosocial and learning behaviors in kindergarten. The sample included 95 mother–father–child triads (child mean age 5.9 years) and 42 kindergarten teachers. All adults reported on their relationship with the child and on their perceptions of parent–teacher relationships. Teachers reported on the child’s behaviors. Main findings: (1) All proximal relationships and the teachers’ relationships with mothers and fathers were related to children’s outcomes; and (2) different patterns of associations were found between father–child and mother–child relationships, and teacher–child relationship, parent–teacher relationships, and children’s outcomes. These findings hint to the different roles of fathers and mothers in their children’s development and to distinguished patterns of relationships of mothers and fathers with kindergarten teachers.

## 1. Introduction

The home and the educational ecosystems are described as two overlapping spheres of influence that share mutual responsibility for children’s social and academic development [1]. Understanding the quality of children’s relationship with their parents and teachers within these systems, as well as the quality of the interactions between parents and teachers, is central to a more complete understanding of developmental pathways in early childhood. Consequently, the main purpose of the current study was to examine the network of child–mother–father–kindergarten-teacher relationships and their association with the child’s prosocial behavior and approach to learning in kindergarten.

### 1.1. Theoretical Framework: Ecological Systems Theory, Networked Model, and Attachment Theory

The study is framed within three theoretical models. First, ecological system theory [2] provides a broad developmental framework that conceptualizes children’s development within six nested systems, defined by their proximity to the child. We focus here on two of these systems: the microsystem, which refers to direct influences such as the direct interaction he/she experiences within the family and kindergarten settings; and the mesosystem, which refers, among others, to the reciprocal relations between systems such as the home and school. The quality of the relationships within and between systems creates a dynamic network of relationships that may support or impair children’s development and adaptation in school [3]. The dynamic systems framework expands the hierarchical additive ecological model and suggests that factors on different levels are fully reciprocal, bidirectional, and synergistic [4], and that self-organization of variables into a complex network of interactions may imply structured patterns [5].

The second organizing theoretical framework is the relatively newly proposed Networked Model [6], which reformulates EST by focusing primary attention on patterns of social interaction and propose the term “setting” to define a set of people that engaged in social interactions. The model suggests that the ecological environment is an overlapping arrangement of structures, each connected to the others by direct and indirect social interactions. Thus, modeling and conducting an analysis of dynamic social networks enable us to have a better understanding of the specific network’s structures [6].

The third theoretical model guiding this study is attachment theory [7], which describes how mutual interactions between children and their main caregivers are generalized into internal working models (IWMs) of attachment. These models guide children’s expectations and behaviors in new situations and, thus, should guide their interactions with their teachers, as well as their socialization processes with peers [8] and academic learning processes [9].

### 1.2. Mother–Child and Father–Child Relationship: Two Unique Relationship Systems?

There is an ongoing interest in the relative and unique contributions of mother–child and father–child relationships to the child’s development. Most studies focused on the important role of the mother–child relationship in the child’s adjustment to school, but in the last several decades, as the father’s involvement in childrearing increases gradually, it has become evident that the father–child relationships contribute in unique ways to the child’s development [10]. Although both parents can be equally sensitive and responsive, mothers and fathers tend to be involved with their children in different ways [4,11]. Some studies show the unique contribution of fathers and indicate that paternal sensitivity is related to better cognitive skills, learning functioning, executive functioning, regulation skills, and reduced externalizing problems [12]. Such effects are moderated by child factors such as age and gender. For example, fathers tend to become more involved during preschool, which is characterized by an increase in behavioral problems [10]. In addition, the father’s—but not the mother’s—negative perception of the relationship with the child is associated with children’s learning and academic difficulties in kindergarten [13], as well as with problem behaviors [14]. However, the current literature is still rather inconsistent about the unique contributions of mothers and fathers to the child’s development [15,16]. Thus, more research on the unique and mutual contributions of both parents in two-parent traditional families to child outcomes is still very much needed.

### 1.3. Parent–Teacher Relationship

In recent decades, the discourse on family–school partnerships underscored the quality of parent–teacher relationships [1,17]. Parents’ involvement in their child’s education is a multidimensional concept that is influenced by various demographic and personal factors of both the parents and teachers, and there are indications that more parental involvement in early childhood is positively related to children’s school adjustment [18,19]. The ability to create meaningful parent–teacher partnerships is strongly related to the quality of the teacher–child relationship [20], as well as to better child adjustment [21,22]. In addition, better parent–child relationships were found to relate to preschool teachers’ more positive perceptions of their relationships with mothers and fathers [23].

Parent–teacher relationships may be even more crucial in early childhood because young children are more dependent and vulnerable, and parents, accordingly, may wish to advance the development of trust and reliance with the educators [24] to protect their offspring. Whereas the parent–teacher relationships were examined quite extensively with respect to the school years, relatively little is known about the role of the parent–teacher relationship in predicting children’s preschool outcomes in the early years. Moreover, to the best of our knowledge, there are no previous studies that examined the differences between mother–teacher and father–teacher relationships during these formative years and their associations with children’s behavioral outcomes.

### 1.4. Expected Outcomes: Prosocial and Learning Behaviors

We focus here on two behavioral outcomes that are deemed crucial for school readiness: prosocial behaviors and positive learning behaviors. Prosocial behavior refers to positive social interactions with others, such as cooperating, helping, sharing, and comforting, and has been found to be a strong predictor of academic development [25]. In contrast, low levels of social competence and higher problem behaviors in preschool are associated with difficulties with pre-academic abilities and influence adaptation and school readiness [26,27]. Prosocial behaviors are linked to the quality of young children’s relationships with mothers, fathers [8,28], and kindergarten teachers [11]. Early positive learning behaviors, typically described as more positive “approaches towards learning”, are linked to the quality of the parent–child relationship [29], the quality of the teacher–child relationship [30], and home–school collaborations [31] and are thus considered crucial for school adaptation.

### 1.5. The Current Study

The current study examined the associations between the mother–father–teacher–child relationship network and the child’s social and learning behaviors in kindergarten. We measured three proximal relationships with the child (mothers, fathers, and teachers) and four perspectives on parent–teacher relationships (mother–teacher, father–teacher, teacher–mother, and teacher–father). Based on the literature review above, our hypotheses were as follows: **(H1)** The quality of the parent–child relationship will be positively related to the quality of the teacher–child relationship. **(H2)** The quality of all adult–child relationships will be positively related to better social and learning skills. **(H3)** The quality of all adult–child relationships will be positively related to better parent–teacher relationships. **(H4)** Positive perceptions of teachers and parents of their mutual relationships will be related to children’s better social and learning performance in kindergarten. **(H5)** Perceptions of teachers and parents of their mutual relationships will be associated. **(H6)** Finally, based on the Networked Model, we expected that some aspects in the mothers’ relationships network will mediate the associations between father–teacher relationships and children’s outcomes and that some aspects in the fathers’ relationships network will mediate the association between mother–teacher relationships and children’s outcomes.

Figure 1 presents the conceptual model that guided this study.

## 2. Materials and Methods

### 2.1. Participants and Procedure

The sample comprised 95 Israeli child–father–mother triads (children aged 55–81 months, *M* = 68.65 (5.9 years), *SD* = 5.87, 41 boys), as well as all 42 leading kindergarten teachers. Teachers were contacted after receiving approval for the study’s protocol from the University’s IRB (approval 464/16), as well as from the Chief Scientist Office at the Israeli Ministry of Education (approval 9312). Families were recruited using flyers and e-mails distributed by the teachers. Before the beginning of the study, both parents signed a consent form approving their participation in the study. Based on the parents’ reports, the sample was approximately normative in terms of its sociodemographic indicators: The median monthly household income was slightly above average, and approximately 67% of the mothers and 49% of the fathers had an academic degree. Data were collected in 2017–2019 in the northern part of Israel. Both mothers and fathers from the same family completed questionnaires pertaining to parent–child relational quality, parent–teacher relationships quality, family demographics, and other background variables. Teachers completed questionnaires pertaining to the quality of their relationship with each parent, the quality of their relationship with the child, and the child’s prosocial and learning behaviors. Families and teachers received gift cards for their participation.

### 2.2. Measures

#### 2.2.1. Parental–Teacher Relationship

The *Quality and Trust in Parent–Teacher Relationship* questionnaire is based on two existing questionnaires: the *Home–School Relationship* scale [32], which measures the quality of parent–teacher relationships, and the *Family–School Relationship* scale [33], which measures trust in those relationships. Three total scores were produced: (1) **Quality** includes various aspects related to the quality of the relationship and communication between the partners, for example, clarity of communication, consent, support, and sharing. Each of the seven items used a 4-point scale. A sample question for parents reads as follows: “How would you describe the emotional tone of the relationship with your child’s kindergarten teacher?”. (2) **Trust** has two almost identical versions, one for teachers (17 items) and one for parents (19 items). All items begin with a positive statement, e.g., “I am confident that the teacher/mother/father…”, and continues with sentences that reflect a variety of behaviors and beliefs that relate to feelings of security, trust, and respect in relationships (adult–adult and adult–child) and the partner’s ability to promote the child’s development in the context of the educational setting. A sample item for parents reads as follows: “I am confident that the teacher keeps me aware of all the information I need related to school”. A sample item for teachers reads as follows: “I am confident that the parent makes me aware of all the information I need about his/her child”. (3) **Positive total score** (sum of quality and trust) is the third score. In the current study, the Alpha reliabilities for quality, trust, and positive total score were as follows: mothers with teachers—0.84, 0.96, and 0.94; fathers with teachers—0.82, 0.94, and 0.95; teachers with mothers—0.74, 0.96, and 0.94; and teachers with fathers—0.84, 0.96, and 0.95, respectively.

#### 2.2.2. Adult–Child Relationship

Two very similar scales were used to measure adults’ perception of their relationship with a particular child: the *Student–Teacher Relationship Scale* [34] (28-items: STRS; Pianta, 2001) and the *Child–Parent Relationship Scale* [35] (30-items; CPRS; Pianta, 1992). Both scales measure patterns of relationships in terms of closeness (“I share an affectionate, warm relationship with this/my child”), conflict (“this/my child and I always seem to be struggling with each other”), and dependency (“this/my child reacts strongly to separation from me”). Alpha reliability scores of the scales were as follows for teachers, mothers, and fathers: closeness = 0.83, 0.64, and 0.81; conflict = 0.76, 0.82, and 0.81; and dependence = 0.59, 0.57, and 0.58, respectively.

#### 2.2.3. Teacher Rating of Children’s Behavior in Kindergarten

**Prosocial skills** were assessed using the prosocial behavior scale of *The Strengths and Difficulties Questionnaire*, teacher version [36] (SDQ; Goodman, 1997). The SDQ is a short behavioral screening questionnaire designed to assess the social–emotional behavior and adaptation to school of children aged 4–16 years. The prosocial scale used here contains five items (sample item: “considerate of other children’s feelings”). Each item is scored on a 3-point Likert scale. Alpha reliability in the current study was 0.77.

**Approaches to learning** were assessed using the *Preschool Learning Behaviors Scale* [37] (29-items; PLBS; McDermott et al., 2002). The PLBS is a teacher-reported questionnaire that measures observable behaviors related to learning and is divided into three main dimensions: competence/motivation, attention/persistence, and attitude toward learning. Each item is scored on a 3-point Likert scale. In the current study, we used a total score of “**positive learning behavior**” generated from summing all items (reversing the negative items). Alpha reliability in the current study for this scale was 0.87.

#### 2.2.4. Control and Background Variables

Parents provided information on household income, parental education, and other sociodemographic characteristics. In addition they completed two questionnaires: (1) *Parent Sense of Competence Scale* [38] (17 items; PSOC; Johnston and Mash, 1989), which relates to the parent’s feelings of satisfaction with the parental role and self-perception as a parent; and (2) *Active and Passive Parental Involvement at School and at Home Scale* (18-items) [39,40] (Fisher and Friedman, 2003; Conduct problems prevention research group, 1995; PTIQ).

## 3. Results

All analyses were conducted with the SPSS software, version 27 [41]. Preliminary analyses were conducted using simple r-correlations and *t*-tests. The study’s main research questions were analyzed with linear regressions, including all main study variables, as well as the control variables, where applicable.

### 3.1. Preliminary Analyses

Table 1 provides descriptive statistics for all study variables. It also shows correlations between parental involvement and competence and the study’s main variables, and, as can be seen, some of these correlations were significant. Additionally, compared with fathers, mothers reported a more positive perception of the parent–teacher relationship (*t*(84) = 5.22, *p* < 0.001) and higher parental involvement in child’s education (*t*(84) = 9.86, *p* < 0.001); teachers reported more positive perceptions about the teacher–mother relationship compared to the teacher–father relationship (*t*(81) = 5.90, *p* < 0.001); and the father’s education was positively related to positive teacher–father relations (*r* = 0.32, *p* < 0.01). No age or child’s gender differences were found in relation to the study’s outcome variables.

### 3.2. Main Analyses

The findings related to Hypotheses 1–4 are presented in Table 2, below:

**Hypothesis 1.** 
*We found a positive correlation between closeness in the mother–child relationship and closeness in the teacher–child relationship. In addition, conflict and dependency in the father–child relationship were positively related to conflict in the teacher–child relationship.*


**Hypothesis 2.** 
*Closeness in the mother–child and in the teacher–child relationship was positively related to prosocial behaviors. Closeness in the teacher–child relationship also was positively related to learning behaviors. Finally, conflict in the father–child and in the teacher–child relationships were negatively related to the child’s prosocial and learning behaviors.*


**Hypothesis 3.** 
*We found that mother–child closeness was positively related to the teacher–mother relationship, while mother–child conflict was negatively related to the father–teacher relationship. Additionally, higher levels of dependency in the father–child relationship were related to a less positive teacher–father relationship. We also found that closeness in the teacher–child relationship was positively related to the teacher–mother relationship.*


**Hypothesis 4.** 
*No associations were found between mother–teacher and father–teacher relationships and the child’s outcomes. Among teachers, we found that more positive teacher–mother relationships were associated with children’s higher prosocial and better learning behaviors. We also found that trust in the teacher–father relationship was positively related to learning behaviors (r = 0.24; p < 0.05).*


**Hypothesis 5.** 
*Table 3 shows a match between mothers and teachers with respect to trust, but not with respect to quality. Among fathers and teachers, we found no match in quality or trust, although quality in father–teacher relationships was positively related with the teachers’ positive perception of their relationship. In addition, fathers’ and mothers’ perceptions of the parent–teacher relationship were in full accord, as were the teachers’ perceptions of their relationships with the mothers and fathers.*


**Hypothesis 6.** 
*We used the Four Steps Model to examine meditational paths [42]. We found no evidence that any of the mother’s or father’s aspects assessed in this study mediated the link between the other parent’s relationship quality with the teacher and children’s outcomes. Still, other interesting mediation paths were found and exemplify the complexity of the associations found here: First, the association between closeness in the mother–child relationship and the child’s prosocial behavior was found to be fully mediated by the teacher’s perception of his/her relationship with the mother as positive (Figure 2).*


Second, the association between closeness in the mother–child relationship and closeness in the teacher–child relationship was found to be fully mediated by the teacher’s perception of his/her relationship with the mother as positive (Figure 3).

Third, the association between the mother’s and teacher’s perceptions of their relationship as positive was fully mediated by the mother’s parental involvement (Figure 4).

Fourth, the association between the father’s sense of competence and the father’s perception of his relationship with the teacher as positive was fully mediated by the level of conflict in the mother–child relationship (Figure 5).

Fifth, the association between fathers’ parental involvement and the father’s perception of his relationship with the teacher as positive was partly mediated by mother–child conflict (Figure 6).

## 4. Discussion

The study examined a relationship network surrounding the child and the relative contribution of each dyadic relationship to the child’s academic and social functioning in kindergarten. The results indicate that all proximal adult–child relationships and the teacher’s perceptions of relationships with mothers and fathers were related to the child’s social and learning behaviors. These findings support previous studies that demonstrated these developmental paths [8,11,23,28,29,30]. In addition, in our study, both parents’ perceptions of relationships with the teacher were not related to the child’s outcomes, similarly to the findings of a recent study related to toddlers’ socioemotional functioning [23].

### 4.1. Mothers and Fathers: Differences in Associations with Teacher–Child Relationship, Parent–Teacher Relationships, and Child’s Adjustment in Kindergarten

Our findings indicate that different dimensions of the mother–child and father–child relationships are related to the quality of the child’s interactions with the teacher, to the father–teacher relationship, to the teachers’ perceptions of relationships with fathers and mothers, and to the child’s social and academic behaviors. These findings support the notion that relationships that occur within one ecosystem (the family) are associated with those within other ecosystems (in this case, the school) [6]. Additionally, our findings suggest that mothers and fathers operate in additive, complementary, and synergistic ways but that their contributions to the child’s behavior, as well as their relationship with the child teacher, are not similar. For example, we found that more experiences of closeness, affection, open communication, and warmth in the mother–child dyad and fewer experiences of anger and conflicts in the father–child dyad are associated with the child’s social and learning behaviors. Thus, our findings contribute to the discourse about similarities and differences between fathers and mothers both in relation to the type of relationships they maintain and in relation to their contribution to their children’s development [4]. These differences may be the result of different parenting styles, as well as of possible different social and cultural expectations from fathers and mothers.

With respect to the teacher–child relationship, the current study points to the connection and continuity between the family and the kindergarten. Previous research suggests that the child’s affection toward his/her mother relates to affectionate teacher–child relationship, whereas child negativity with the mother relates to dependency in the teacher–child relationship [43]. Our findings add to this by emphasizing both parents’ contribution to the child’s better relationship with his/her teacher.

In addition, we found that more closeness in the mother–child relationship relates to a more positive teacher–mother relationship and, on the other hand, that a less conflictual mother–child relationship relates to a more positive father–teacher relationship. A different pattern of relations was found in the fathers’ case, where more dependency in the father–child relationship was related to a less positive teacher–father relationship. These findings show that parent–child relationships are associated with adults’ perceptions of parent–teacher relationships and, moreover, that each parent has a unique contribution.

As stated, there is ample evidence supporting the assumption that mother–child and father–child relationships contribute in different ways to a child’s development. Studies indicate that mothers and fathers may have different parental styles (physical, social, or cognitive) during nurturance, play, and learning situations and may use different co-regulation interactions when the child is in distress [8,31]. While maternal closeness appears to be a strong predictor of child adjustment [44], Activation Relationship Theory (ART) [45] suggests that fathers may influence their children’s behavior through different means than mothers. For example, fathers tend to excite and encourage their children to take risks, explore, take chances, and stand up for themselves. They also tend to engage in physical play with their children. ART complements attachment theory in suggesting that, although both parents can be sensitive and ensure the child’s protection, mothers serve more as a “safe haven” for comfort and support in time of distress, and fathers serve more as a “secure base” for promoting the child’s outside world exploration, emotion regulation, and social skills [16,45]. The current study’s findings do not directly support these suggestions but may provide an indirect look into the differences in mothers and fathers’ perceptions of important factors in their relationships with the child and with the teacher. Among mothers, closeness was associated with children’s adjustment, whereas, among fathers, conflict and dependence were the more salient predictors of different relational and adjustment factors. These findings also support previous research studies showing that, in two-parent traditional families, mothers and fathers advance and regulate children’s social and pre-academic behaviors in diverse ways [12,13,14], and that the effects of the proximal parent–child interaction extend to the parent–teacher relationships.

### 4.2. Mothers and Fathers: Different Subsystems in Parent–Teacher Relationships

A snapshot view of the current findings about the mother–father–teacher relationships shows diverse patterns of variable groups [5], reflecting the fact that the perceptions of mothers, fathers, and teachers regarding the parent–teacher relationship relate to different sets of variables and connections.

#### 4.2.1. Mother-and-Teacher Subsystem

Consistent with the research hypotheses, our findings demonstrate that teachers’ perceptions regarding their relationships with mothers are positively related to better mother–child and teacher–child relationships and to the better adjustment of the child [46,47]. We also found that teachers perceived their relationship with the mothers more positively compared to their relationship with fathers. Accordingly, mothers perceived their relationship with teachers more positively than fathers did. Contrary to our hypotheses, the mother’s perception of her relationship with the teacher did not relate to her relationship with the child or to teacher–child relationships, nor to any of the child’s behavioral outcomes. However, we found that the mother’s involvement in her child’s education fully mediated the association between the mother’s positive perception of her relationship with the teacher and the teacher’s positive perception of his/her relationship with the mother. Additionally, the findings demonstrate the important role of the teacher’s perspective of his/her relationship with the mother. His/her positive view of these relationships was found to mediate the association between mother–child closeness and teacher–child closeness, as well as the association between mother–child closeness and the child’s prosocial behavior in kindergarten. These findings provide insight into the type of mechanisms that enables the continuity and overlapping between the family and the educational settings.

#### 4.2.2. Father-and-Teacher Subsystem

The patterns found in relation to the father-and-teacher relationships were different from those described for mothers and teachers. We found that teachers’ more positive perception of relationship with fathers was related to less dependency reported in the father–child relationship and that more trust in these relationships was associated with better learning behaviors for the child. Interestingly, we found that a higher level of the mother’s sense of satisfaction and confidence in her parental role increased both the father’s and teacher’s positive perceptions of their mutual relationship. Additionally, higher parental education of both parents was associated with a more positive teacher’s perception of his/her relationship with the father. Unlike the teacher–mother relationships, we found no accordance between fathers’ and teachers’ perceptions of their mutual relationships.

The role of the father–teacher relationships could be better understood from other associations found in this study. For example, we found that higher levels of fathers’ sense of competence and involvement were associated with better father–teacher relationships and that more conflicts in the mother–child dyad was associated with less positive father–teacher relationships. In addition, the mediating role we found of mother–child conflict in the associations between fathers’ competence and involvement and positive father–teacher relationships may indicate that mother–child conflict may alter fathers’ ability to develop positive communication and trust with teachers. It seems that, compared to the mother–teacher relationship, which seems to not be affected by paternal aspects, the father–teacher relationship is dependent in many ways on aspects relating to the mother.

### 4.3. Limitations

There are several limitations that should be considered in relation to the findings reported here. First, the relatively small sample size limited the ability to detect significant effects. Furthermore, the sample size may affect the ability to generalize and make strong conclusions. This should be addressed in future studies with larger samples. Second, data collection was based on questionnaires and interviews and did not include direct observations. Third, the study involved cross-sectional data, so causal inferences are not possible. Accordingly, the generalization of these findings should be made with caution. Fourth, the study included only traditional two-parent families, whereas the number of non-traditional two-parent families (e.g., same-sex parents) is growing and should be examined as well. Finally, the study did not include important factors that could potentially moderate our findings, such as child temperament and aspects related to family dynamics. Future studies with larger samples, longitudinal designs, and more diverse families and measurement batteries are needed to deepen our understanding of the study findings and to expand our knowledge about the relationship networks associated with child development.

## 5. Conclusions

This study promotes an understanding of the distinction and overlap between relationship networks within the family and in the educational settings and their relation to the child’s adjustment. The results provide information on the unique and shared contributions of mothers, fathers, and teachers to the child’s development and reveal the patterns of two subsystems of relationships: mothers–teachers and fathers–teachers. This study confirms the need to expand the multilevel perspective and advance the knowledge which promotes positive family–school partnerships and desirable development in early childhood.

## Figures and Tables

**Figure 1 children-10-01102-f001:**
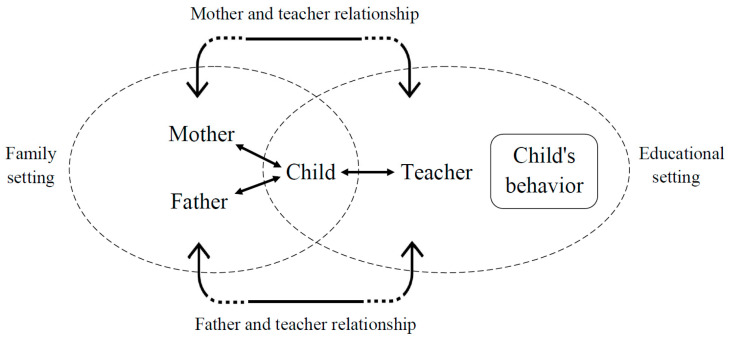
The study’s conceptual model: adult–child interaction = microsystemic relationships, and parent–teacher interactions = mesosystemic relationships. Based on Neal & Neal, 2013.

**Figure 2 children-10-01102-f002:**
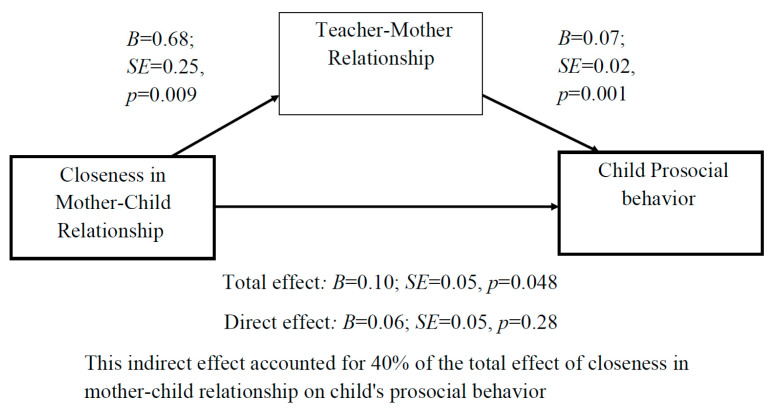
Mediation analysis: The association between closeness in mother–child relationship and the child’s prosocial behaviors is fully mediated by the teacher–mother relationship.

**Figure 3 children-10-01102-f003:**
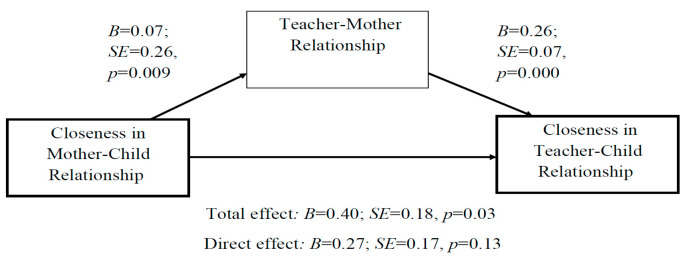
Mediation analysis: The association between closeness in the mother–child relationship and closeness in the teacher–child relationship is fully mediated by the teacher–mother relationship.

**Figure 4 children-10-01102-f004:**
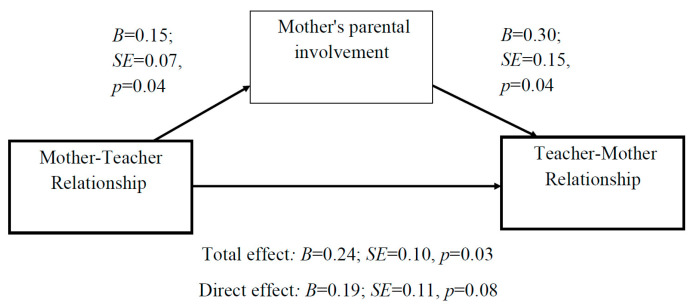
Mediation analysis: The association between the mother–teacher relationship and the teacher–mother relationship is fully mediated by the mother’s parental involvement.

**Figure 5 children-10-01102-f005:**
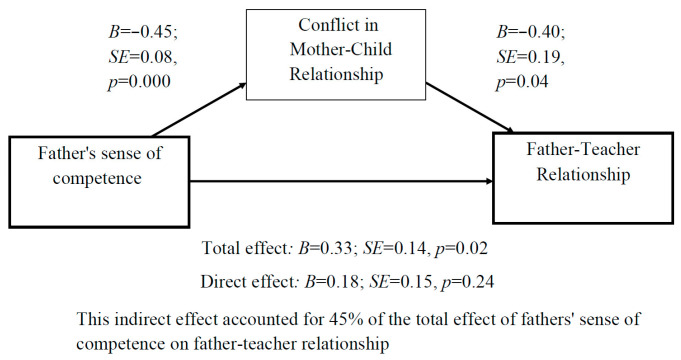
Mediation analysis: The association between fathers’ sense of competence and the father–teacher relationship is fully mediated by the mother–child conflict.

**Figure 6 children-10-01102-f006:**
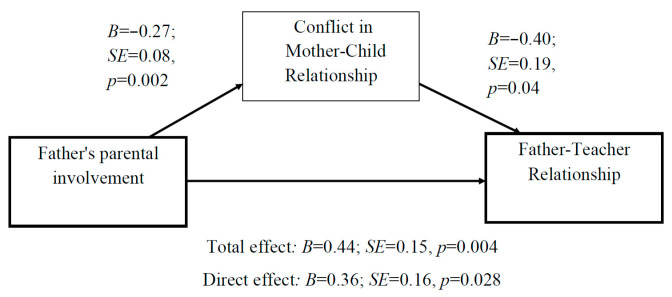
Mediation analysis: The association between fathers’ parental involvement and the father–teacher relationship is partially mediated by the mother–child conflict.

**Table 1 children-10-01102-t001:** Means and *SDs* for study variables, and correlation matrix for control variables.

	Possible Min–Max	Actual Min–Max	*M*	*SD*	M-PSOC	F-PSOC	M-IP	F-IP
**M-C Relationship**								
M-C Closeness	11–55	35–54	45.41	4.17	0.35 **	0.21	0.36 **	0.30 **
M-C Conflict	13–65	11–50	26.02	7.36	−0.49 ***	−0.35 **	−0.22 *	−0.32 **
M-C Dependence	6–30	7–26	15.40	3.80	−0.19	−0.11	−0.03	−0.11
**F-C Relationship**								
F-C Closeness	11–55	19–54	43.11	5.93	0.21 *	0.46 ***	0.24 *	0.28 **
F-C Conflict	13–65	4–47	26.40	8.02	−0.21 *	−0.49 ***	−0.12	−0.25 **
F-C Dependence	6–30	6–24	15.54	3.80	−0.09	0.04	0.02	−0.10
**T-C Relationship**								
T-C Closeness	11–55	14–55	41.75	7.21	0.12	0.03	0.08	0.09
T-C Conflict	12–60	3–38	15.44	5.01	−0.11	−0.16	−0.32 **	−0.30 **
T-C Dependence	5–25	5–21	8.15	2.82	−0.01	0.02	−0.08	−0.13

**Mother with Teacher**	078	33–82	70.57	10.16	0.17	−0.05	0.21 *	0.04
**Father with Teacher**	0–78	20–81	63.86	13.32	0.32 **	0.24 *	0.16	0.30 **
**Teacher with Mother**	0–72	22–77	65.67	10.22	0.12	0.10	0.26 *	0.09
**Teacher with Father**	0–72	3–77	59.09	12.31	0.25 *	0.02	0.12	0.08

**Learning Behavior**	0–58	11–58	46.87	9.03	0.11	0.20	0.14	0.16
**Prosocial Behavior**	0–10	1–10	7.74	2.14	0.08	0.15	0.23 *	0.10

M-PSOC	17–102	54–92	75.60	8.22	−	0.50 ***	0.24 *	0.18
F-PSOC	17–102	39–92	74.62	9.96	0.50 ***	−	0.34 ***	0.38 ***
M-PI	0–72	25–66	46.33	7.08	0.24 *	0.34 ***	−	0.52 ***
F-PI	0–72	11–67	38.28	9.13	0.18	0.38 ***	0.52 ***	−

* *p* < 0.05; ** *p* < 0.01; *** *p* < 0.001. *Note*: M = mother; F = father; C = child; T = teacher; PSOC = Parent Sense of Competence; PI = parental involvement.

**Table 2 children-10-01102-t002:** Correlations matrix for study variables.

	T-C Relationship			C Behaviors		Parent-and-Teacher Relationships			
	T-C Closeness	T-C Conflict	T-C Dependence	PLBS	Prosocial	M-T-R	F-T-R	T-M-R	T-F-R
**M-C Relationship**									
M-C Closeness	0.23 *	−0.07	0.13	0.18	0.21 *	0.11	0.14	0.28 **	0.21
M-C Conflict	0.12	0.12	−0.11	0.00	−0.07	−0.08	−0.22 *	0.00	0.06
M-C Dependence	0.14	0.02	−0.00	0.03	0.06	0.03	−0.06	0.07	−0.05
**F-C Relationship**									
F-C Closeness	0.05	0.01	0.03	0.17	0.08	0.07	0.12	0.11	−0.04
F-C Conflict	0.01	0.24 *	−0.02	−0.25 *	−0.27 **	0.19	0.07	−0.04	−0.11
F-C Dependence	0.08	0.21 *	0.11	−0.10	−0.30	0.01	0.06	0.00	−0.28 *
**T-C Relationship**									
T-C Closeness	−	−	−	0.35 **	0.26 *	0.13	−0.08	0.37 ***	0.15
T-C Conflict	0.04	−	−	−0.24 *	−0.49 ***	0.12	0.17	−0.14	−0.04
T-C Dependence	0.01	0.48 ***	−	−0.12	−0.07	0.14	−0.03	−0.04	−0.05

PLBS	0.35 **	−0.24 *	−0.12	−	0.38 ***	−0.06	−0.15	0.28 **	0.10
Prosocial	0.26 *	−0.49 ***	−0.07	0.38 ***	−	0.03	−0.19	0.33 **	0.12

** p* < 0.05; ** *p* < 0.01; *** *p* < 0.001. *Note*: M = mother; F = father; C = child; T = teacher; R = relationship; PLBS = Preschool Learning Behaviors Scale. For the parent–teacher relationship score, only the total positive score is presented (associations with trust and quality, if they existed, are presented in the text).

**Table 3 children-10-01102-t003:** Correlations matrix for parent–teacher relationship variables.

	Mother–Teacher Relationship			Teacher–Father Relationship		
	M-T Quality	M-T Trust	M-T Total	T-F Quality	T-F Trust	T-F Total
**Father–Teacher Relationship**						
F-T Quality	0.45 ***	0.38 ***	0.42 ***	0.22	0.13	0.24 *
F-T Trust	0.43 ***	0.54 ***	0.54 ***	0.22	0.10	0.21
F-T Total	0.34 **	0.44 ***	0.43 ***	0.17	0.09	0.19
**Teacher–Mother Relationship**						
T-M Quality	0.17	0.14	0.15	0.33 **	0.21	0.22 *
T-M Trust	0.27 *	0.26 *	0.29 **	0.09	0.58 ***	0.41 ***
T-M Total	0.21 *	0.22 *	0.24 *	0.17	0.55 ***	0.41 ***

* *p* < 0.05; ** *p* < 0.01; *** *p* < 0.001. *Note*: M = mother; F = father; T = teacher.

## Data Availability

Full dataset is available from the corresponding author.

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
