# Peer review of "The Child–Mother–Father–Teacher Relationship Network in Kindergarten and Its Association with Children’s Social and Academic Development: An Ecological Perspective"

_children, 2023, doi:10.3390/children10071102_

Round 1
Reviewer 1 Report
Thank you very much for sharing this paper with me. This is an important paper and I have several minor comments.
The sample size is relatively small. How does this affect your conclusions and their generalizability?
How did you incorporate the cultural and social system of this study context?
Why did you rely on cross-sectional data? Why not longitudinal data or qualitative data?
Is it important to explore potential moderating variables, such as child temperament or family dynamics, could further enhance our understanding of the complexities involved in these relationships?
Author Response
Reviewer 1:
- Thank you very much for sharing this paper with me. This is an important paper and I have several minor comments.
Thank you!
- The sample size is relatively small. How does this affect your conclusions and their generalizability?
We address this important point in the limitations section (p. 12).
- How did you incorporate the cultural and social system of this study context?
The study was conducted in Israel and included a fairly diverse population of Israeli triads and teachers. Although small, the sample was quite representative of Israeli population in large in terms of socioeconomic status, education level, and ethnicity. These aspects are represented in the description of the population as well as in our analyses (in terms of controlling from demographic aspects).
- Why did you rely on cross-sectional data? Why not longitudinal data or qualitative data?
This is an important limitation, and we acknowledge it in the limitation section (p. 12). Unfortunately, this limitation can only be acknowledged but cannot be changed. In the same section, we also describe future studies that should address our research questions using a longitudinal design. As for qualitative analysis, while important, we are not sure why it is relevant to the current, all quantitative study.
- Is it important to explore potential moderating variables, such as child temperament or family dynamics, could further enhance our understanding of the complexities involved in these relationships?
We agree. We controlled for all background variables that were collected in this study, including: household income, parental education, other sociodemographic characteristics, parental Sense of Competence parental satisfaction with the parental, parental self-perception and parental actual involvement in the school (see p. 5). Unfortunately, we cannot explore important variables such as those proposed by the reviewer that were not included in data collection. We added an acknowledgment to this issue to limitation section (p. 12).
Reviewer 2 Report
The paper focuses on child-mother-father-teacher relationship network in kindergarten and its association with children's social and academic development. The special emphasis on father's involvement in childrearing and its effect on child development is a strong advantage of the study since research show an increased participation of fathers in childcare that needs more detail investigation. Another very interesting moment in the papers relates to the emphasis on parent-teacher relationship in predicting children's preschool outcomes. Early child development and the involvement of parents, teachers and professionals from educational, health and social sphere requires a multidimensional analytical approach which is demonstrated very well in the paper. My suggestions to the authors are the following:
1. The authors should specify the methods used in the analyses and the modeling procedures in a separate paragraph in the methods section – correlation, mediation analysis, etc.
2. The results show that “… more experiences of closeness, affection, open communication, and warmth in the mother-child dyad, and fewer experiences of anger and conflicts in the father-child dyad are associated with the child's social and learning behaviors”. This finding could be linked to the parenting styles and explained in the context of the differences between mothers and fathers parent-child communication.
3. I would suggest also to add in the discussion part of the paper more explanations of the mechanisms through which certain relationships take place, e.g. it was found that “…mother-child relationship relates to more positive teacher-mother relationship”; “…more dependency in the father-child relationship was related to less positive teacher-father relationship”. The authors should add more explanations and interpretation of the uncovered empirical dependencies/ relationships.
4. It is not clear which are the control covariates in the mediation models. The control covariates, in any, should be described as well. In the ‘Discussion’ part of the paper it mentioned that “…parental education, of both parents, was associated with a more positive teacher's perception of her relationship with the father”. Is the parental education includes in the multivariate analyses and what other covariates were also controlled for?
5. The period of data collection should be mentioned in the description of the study.
6. I would suggest also moving the section on the limitations of the study after the conclusion and also to extend the conclusions by mentioning the main results/ the most significant from the study.
Author Response
Reviewer 2:
- The authors should specify the methods used in the analyses and the modeling procedures in a separate paragraph in the methods section – correlation, mediation analysis, etc.
As requested, we added the following paragraph at the beginning of the results section (p. 6): “All analyses were conducted with the SPSS software, version 27 [41]. Preliminary analyses were conducted using simple r correlations and t-tests. The study’s main research questions were analyzed with linear regressions including all main study variables as well as the control variables where applicable.”
- The results show that “… more experiences of closeness, affection, open communication, and warmth in the mother-child dyad, and fewer experiences of anger and conflicts in the father-child dyad are associated with the child's social and learning behaviors”. This finding could be linked to the parenting styles and explained in the context of the differences between mothers and fathers parent-child communication.
- I would suggest also to add in the discussion part of the paper more explanations of the mechanisms through which certain relationships take place, e.g. it was found that “…mother-child relationship relates to more positive teacher-mother relationship”; “…more dependency in the father-child relationship was related to less positive teacher-father relationship”. The authors should add more explanations and interpretation of the uncovered empirical dependencies/ relationships.
- It is not clear which are the control covariates in the mediation models. The control covariates, in any, should be described as well. In the ‘Discussion’ part of the paper it mentioned that “…parental education, of both parents, was associated with a more positive teacher's perception of her relationship with the father”. Is the parental education includes in the multivariate analyses and what other covariates were also controlled for?
As mentioned at the beginning of the results section, control variables were included in the main analyses, when applicable (p. 6). This means that whenever a control variable was found related to one of the main study variables, it was included in the regression analysis that included this variable. For example, the father's education was positively related to positive teacher-father relations. Thus, father’s education was entered in all analyses in which the quality of the father-teacher relationships was included.
- The period of data collection should be mentioned in the description of the study.
Data collection period is stated in the “participants and procedures" section on p. 4: “Data were collected in 2017-2019 in the northern part of Israel.
- I would suggest also moving the section on the limitations of the study after the conclusion and also to extend the conclusions by mentioning the main results/ the most significant from the study.
We followed the journal template in instructions for submission that the conclusion section should be the last and final part of the manuscript.